# Impact of pulmonary rehabilitation in sleep in COPD patients measured by actigraphy

**Suman B. Thapamagar**[1,2,3]*, **Kathleen Ellstrom**[2], **James D. Anholm**[1,2], **Ramiz A. Fargo**[1,2,3], **Nagamani Dandamudi**[1,2]

**1** Division of Pulmonary, Critical Care, Sleep, Allergy and Hyperbaric Medicine, Loma Linda University School of Medicine. Loma Linda, CA, United States of America, **2** Pulmonary and Critical Care Section, Medical Services, VA Loma Linda Healthcare Systems, Loma Linda, CA, United States of America, **3** Division of Pulmonary, Critical Care and Hyperbaric Medicine, Riverside University Health Systems, Moreno Valley, CA, United States of America

* suman.thapamagar@ruhealth.org

## Abstract

### Introduction

Chronic obstructive pulmonary disease (COPD) patients have poor sleep quality, longer time to sleep onset and frequent nocturnal awakenings. Poor sleep quality in COPD is associated with poor quality of life (QoL), increased exacerbations and increased mortality. Pulmonary rehabilitation (PR) improves functional status and QoL in COPD but effects on sleep are unclear. PR improves subjective sleep quality but there is paucity of objective actigraphy data. We hypothesized that actigraphy would demonstrate subjective and objective improvement in sleep following PR. Paired comparisons (t-test or Wilcoxon-signed-rank test) were performed before and after PR data on all variables.

### Methods

This retrospective study of COPD patients undergoing PR utilized actigraphy watch recordings before and after 8-weeks of PR to assess changes in sleep variables including total time in bed (TBT), total sleep time (TST), sleep onset latency (SOL), sleep efficiency (SE), wakefulness after sleep onset (WASO) and total nocturnal awakenings. A change in Pittsburg Sleep Quality Index (PSQI) was a secondary outcome. PSQI was performed before and after PR.

### Results

Sixty-nine patients were included in the final analysis. Most participants were male (97%), non-obese (median BMI 27.5, IQR 24.3 to 32.4 kg/m$^2$) with an average age of 69 ± 8 years and 71% had severe COPD (GOLD stage 3 or 4). Prevalence of poor sleep quality (PSQI ≥5) was 86%. Paired comparisons did not show improvement in actigraphic sleep parameters following 8-weeks PR despite improvements in 6-min-walk distance (6MWD, mean improvement 54 m, 95% CI 34 m to 74 m, p<0.0001) and St. George's Respiratory Questionnaire scores (SGRQ, mean improvement 7.7 points, 95% CI 5.2 to 10.2, p<0.0001). Stratified analysis of all sleep variables by severity of COPD, BMI, mood, mental status, 6-MWD and SGRQ did not show significant improvement after PR. In Veterans with poor

**Data Availability Statement:** Data is saved in Loma Linda VA Healthcare Systems server. Data cannot be shared publicly because of the data is the property of VA Healthcare systems. Data are

available from the senior authors or the VA Loma Linda Institutional Data Access / Ethics Committee for researchers who meet the criteria for access to confidential data. IRB Contact: Sunbeam Obomsawin E mail: Sunbeam.obomsawin@va.gov Phone: 909-825-7084 x2264 Senior Author Contact: Nagamani Dandamudi, MD Email: Nagamani.Dandamudi@va.gov.

**Funding:** The author(s) received no specific funding for this work.

**Competing interests:** The authors have declared that no competing interests exist.

sleep quality (PSQI $\geq$ 5), PR improved subjective sleep quality (PSQI, mean difference 0.79, 95% CI 0.07 to 1.40, p = 0.03).

## Conclusions

Pulmonary rehabilitation improved subjective sleep quality in Veterans who had poor sleep quality at the beginning of the PR but did not improve objective sleep parameters by actigraphy. Our findings highlight the complex interactions among COPD, sleep and exercise.

## Background

Adults with Chronic obstructive pulmonary disease (COPD) have poor sleep quality [1–3]. There is an increased prevalence of sleep symptoms such as difficulty of initiating and maintaining sleep or excessive daytime sleepiness among people with COPD [4]. People with severe COPD also have longer time to sleep onset and fragmented sleep due to frequent arousals [5]. Severity of COPD is also associated with poor sleep quality [6]. Additionally, poor sleep quality in people with COPD is associated with poor health related quality of life [1, 2], increased frequency of COPD exacerbations and mortality [7]. Poor sleep may also alter immune function and increase likelihood of COPD exacerbations [8].

Several factors, such as cough and dyspnea, affect sleep in people with COPD [7]. Daytime hypoxemia is independently associated with poor sleep efficiency [5, 9]. Nocturnal desaturation, however, does not have a clear relationship with poor sleep quality [10]. Pharmaceuticals used in the management of COPD such as corticosteroids and beta-agonists can also disrupt sleep and cause poor sleep quality [11].

Pulmonary rehabilitation (PR) has been proven to be beneficial in the management of COPD. It decreases dyspnea, increases exercise tolerance and improves quality of life (QoL) [12–14]. PR may provide short-term mortality benefit as well, if done early after acute exacerbation of COPD [15]. However, there are only sparse data on the impact of PR on sleep. A prospective study by McDonnell and colleagues showed that PR did not improve sleep quality as measured by Pittsburgh Sleep Quality Index (PSQI) despite improvement in QoL, exercise capacity, anxiety and depression [1]. Another study done in the chronic lung disease population by Soler et al. showed significant improvement in sleep quality after PR, even in the sub-group of COPD population [16]. There is also evidence that COPD patients with better sleep measures, such as non-fragmented sleep, good sleep efficiency, long bouts of sleep and short wakefulness after sleep onset, have increased activity during daytime and spent more time in the daylight [6]. If PR can improve sleep measures, improved sleep may, in turn, improve PR related outcomes.

All studies evaluating effects of PR in improving sleep quality in COPD have used subjective measures of sleep quality such as the PSQI. There is paucity of published studies measuring objective sleep measures such as sleep duration, sleep efficiency, nighttime awakening and wake after sleep onset. Therefore, we designed this retrospective, in-group, before-and-after study to evaluate effect of pulmonary rehabilitation in the quality of sleep in people with COPD using both subjective sleep quality (PSQI) and actigraphy data.

## Methods

### Study design and setting

We performed a retrospective "pre-and-post" study of prospectively maintained cohort data of people with COPD who underwent eight weeks of an outpatient center-based PR. This was a

single-center study evaluating the Veteran population in Southern California. The study was approved by the Loma Linda Healthcare Systems institutional review board (IRB– 605, Protocol # 1219). The IRB waived consent requirement since the study was retrospective in design and data was analyzed anonymously.

## Study participants and enrollment criteria

We reviewed charts of all Veterans who participated in PR at the VA Loma Linda Healthcare System from January 1, 2012 through December 31, 2018; and completed before-and-after rehabilitation watch actigraphy for sleep assessment. Only Veterans (older than 18 years of age) who underwent PR for COPD were included in the study. Veterans who did not complete the entire 8-weeks of the rehabilitation program (drop-outs) or had less than three days of actigraphy data, both before and after PR, were excluded from the study. We did not exclude current smoker in the study. Our PR program at one point allowed current smokers to participate. This exception was subsequently stopped due to high dropout rates among smokers. Since then only current non-smokers are allowed to participate in the PR program.

## Pulmonary rehabilitation program

The PR program at VA Loma Linda Healthcare System is a comprehensive American Association of Cardiovascular and Pulmonary Rehabilitation (AACVPR) certified program for Veterans. The program involves exercise training and extensive self-management educational programs including education on COPD symptoms, symptoms management, inhaler techniques, nutritional assessments and interventions and psychosocial evaluations. Participants meet three-days-a-week for eight weeks. Each PR session comprises of 30–60 minutes of educational sessions and 2 to 2.5 hours of exercise sessions with at least 30 minutes session of treadmill exercise tailored to each individual's tolerance to exercise. As part of the PR program each participant completes an extensive list of questionnaires including demographics, medical history, medications, general health survey and various instruments assessing depression, mental status, dyspnea and sleep both before and after the PR.

All participants of the PR program at our center are asked to wear an Actiwatch 2® (Respironics Inc., Murrysville, PA) on the wrist 24-hour a day for at least a week before the program start and after completion of 8-weeks of the PR for a maximum of two weeks. At each time interval, the data is downloaded onto a computer via the Actiwatch 2® communication dock system. The Actiwatch2® software analyses data and a summary output including total time in bed (TBT), total sleep time (TST), sleep onset latency (SOL), sleep efficiency (SE), wakefulness after sleep onset (WASO) and number of nightly awakenings is recorded for each of participant.

## Assessments, data collection and outcomes

An actigraph is small simple device typically worn on the wrist like a watch. It can, however, be worn on an ankle or waist. This device is a sophisticated movement sensor that continuously records movement and integrates it over time. Mathematical algorithms are then utilized to estimate sleep and wakefulness time [17]. Use of actigraphy to monitor nocturnal sleep-wake parameters is feasible and simple. It is an accepted method to measure sleep parameters in clinical practice and is used in research as well [18, 19]. It has been studied in both the COPD and non-COPD populations; and has acceptable validity and sensitivity to detect sleep patterns associated with sleep disorders and medications [6, 19, 20].

Sleep quality was measured using the PSQI which was collected pre- and- post PR. PSQI measures sleep disturbances and sleep habits during last one month period. The questionnaire has 19 individual items and includes open ended questions and Likert-like items. It generates

seven clinically derived domains or components of scores which includes: sleep quality, sleep latency, sleep duration, habitual sleep efficiency, sleep disturbances, use of sleeping medications, and daytime dysfunction. The global PSQI score is calculated by adding scores from all seven domains [21]. PSQI is a self-administered questionnaire with high test-retest reliability. It has been well validated to assess quality of sleep subjectively [21–23]. A global PSQI score of ≥ 5 reliably distinguishes "poor" quality sleepers from "good" quality sleepers (diagnostic sensitivity 89.6% and specificity 86.5%) [21].

We also collected data on COPD related QoL and symptom burden measurements with the COPD assessment test (CAT) [24], Modified Medical Council Research (mMRC) dyspnea scale [25], visual analogue scale part of Pulmonary Functional Status and Dyspnea Questionnaire (PFSDQ) [26] and St. George's respiratory questionnaire (SGRQ) [27]; exercise capacity assessment using 6-minute walk test (6-MWT) [28]; mental status and depression assessments using St. Louis University Mental Status Examination scale (SLUMS) [29] and 30-points Geriatric Depression Scale (GDS) [30]; and daytime sleepiness using the Epworth Sleepiness Scale (ESS) [31]. These measurements were collected both before and after PR.

Data abstraction was done by clinicians involved in the care of the study participants. Most of the data abstracted were from pulmonary rehabilitation chart. In case of missing data, study participants clinical chart was accessed by clinicians directly involved in their care.

The primary outcome measure for this study was changes in sleep variables measured by actigraphy after 8-weeks of PR. Change in PSQI score was a secondary outcome measure.

## Statistical analysis

All subjects who completed 8-weeks of PR program and have at least three days of both pre-PR and post-PR actigraphy data, were included in the final analysis. Shapiro-Wilk tests were performed to assess the normality of the data. Descriptive statistics are presented as mean (± SD) and median (Interquartile range, IQR) where applicable. Paired comparisons with paired t-test and Wilcoxon signed rank test were performed where applicable. Stratification of each of the variables (COPD severity, BMI categories, GDS, SLUMS, 6MWT, SGQR) based on clinically significant difference and paired comparisons were performed. Stratified analyses of all actigraphy sleep variables were performed based on severity of COPD, BMI, mood, mental status, 6-MWT, SGRQ scores and quality of sleep by PSQI. For PSQI, the study groups were stratified into normal quality sleepers (PSQI score of <5) and poor-quality sleepers (PSQI score of ≥5). Stata v15.0 (StataCorp, College Station, TX) was used for statistical analysis. A p-value < 0.05 was considered statistically significant.

## Results

We reviewed medical charts of 90 Veterans who underwent PR to screen for the study. Thirteen participants were excluded because the indication for PR was other than COPD (Fig 1). Seventy-seven participants were identified who underwent PR for COPD. Eight more participants were excluded from the final analysis due to missing pre- or post- rehabilitation actigraphy data. Overall, 69 participants were included in the final analysis. Median number of days of actigraphy use before PR was 5 days (IQR 4–7) and 4 days (IQR 4–5) after PR.

## Demographics and baseline clinical characteristic

The baseline characteristics of the study population are summarized in Table 1. The study population was predominantly male (97%) with mean age of 69 ± 8 years. More than half of the Veterans were not obese (62.3% had BMI less than 30 kg/m$^2$). More than two-thirds (69.5%) of the Veterans had severe or very severe COPD (GOLD stage III or IV) and the impact of

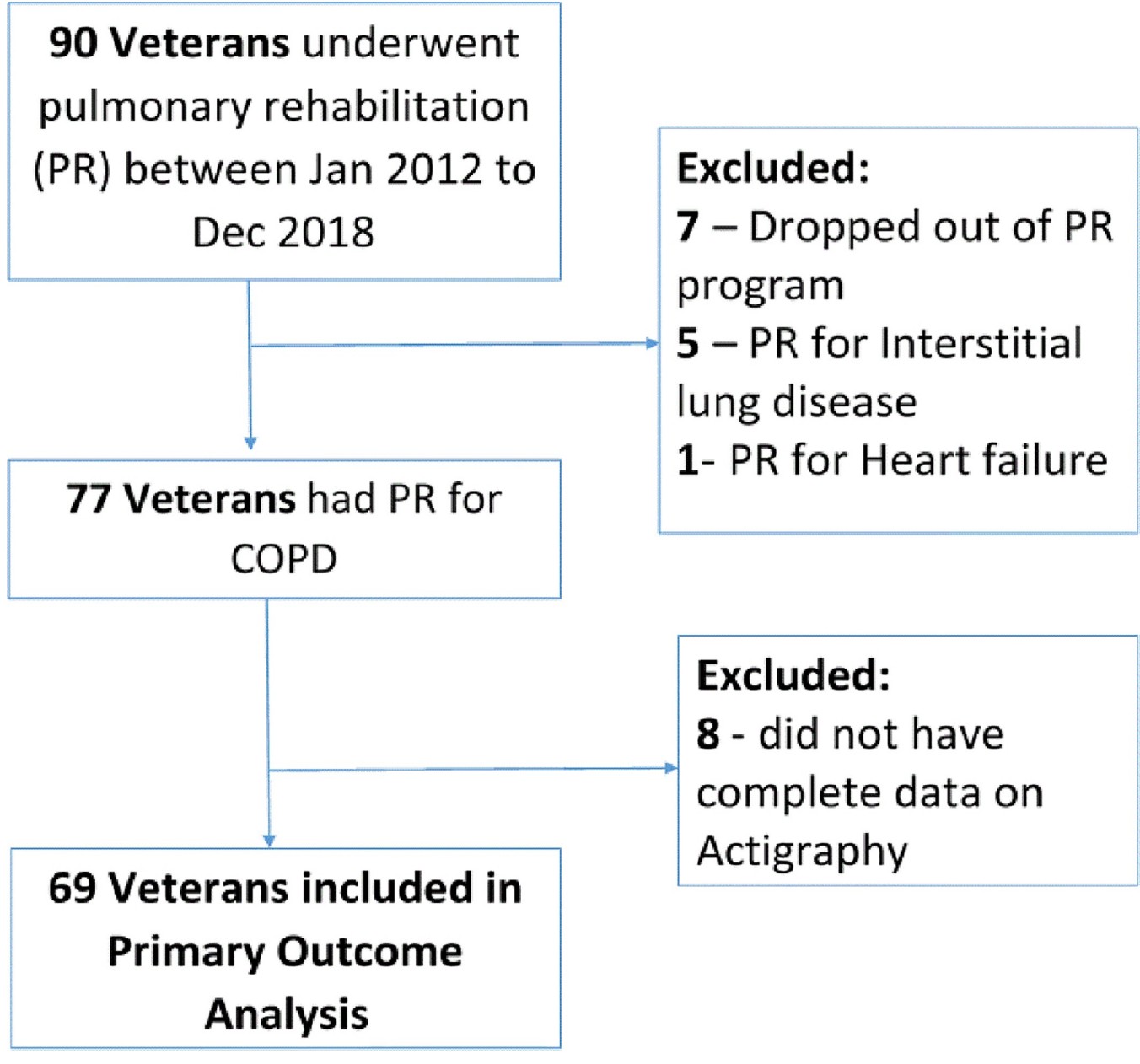

**Fig 1. Study flow chart.**

COPD in their lives as measured by COPD Assessment Test (mean CAT score 24.8 ± 7.1) was very high. Moreover, 75% of the study population had a high or very high impact of COPD on their daily lives. The SGRQ score was also moderately high (median 61.1, IQR 48.4 to 72.5) indicating high burden of COPD. The average dyspnea score (mMRC) was 2.4 ± 1.1. The six-minute walk distance (6-MWD) was relatively preserved (median 373m, IQR 300m – 432m).

### Sleep quality and factors impacting sleep

Poor sleep quality was highly prevalent in our study population. Eighty-six percent of the study population had poor sleep quality, defined by PSQI score of ≥ 5 and all participants had

**Table 1. Baseline characteristics of the study population.**

| Demographics | |
|---|---|
| **Age**, (mean ± SD), years | 69 ± 8 |
| **Male**, n (%) | 67 (97) |
| **BMI**, kg/m$^2$, median (IQR) | 27.5 (24.3–32.4) |
| Non-obese (BMI <30), n (%) | 43 (62.3) |
| Obese (BMI 30–39.9), n (%) | 20 (29.0) |
| Morbidly obese (BMI ≥ 40), n (%) | 6 (8.7) |
| GOLD stages, n (%) | |
| GOLD II | 21(30.4) |
| GOLD III | 33(47.8) |
| GOLD IV | 15(21.7) |
| **FEV$_1$% predicted**, mean ± SD | 47.4 ± 14.9 |
| **Impact of COPD** | |
| **CAT score,** n = 52, mean ± SD | 24.8 ± 7.1 |
| Medium impact—CAT score >10 - ≤ 20, n (%) | 13 (25) |
| High impact—CAT score >20–30, n (%) | 39 (75) |
| **mMRC Dyspnea Scale**, n = 63, mean ± SD | 2.4 ± 1.1 |
| **SGRQ score, n = 67,** median (IQR) | 61.1 (48.8–72.5) |
| **6-min walk test (meters)**, n = 69, median (IQR) | 373 (300–432) |
| **Prevalence of co-morbidities, n (%)** | |
| Hypertension | 43 (62) |
| Dyslipidemia | 40 (58) |
| Coronary artery disease | 24 (35) |
| Diabetes mellitus | 24 (35) |
| **Prevalence of co-morbidities potential effect on sleep**, n (%) | |
| Chronic pain syndrome | 36(52) |
| Psychiatric conditions | 33 (48) |
| Home oxygen | 18(26) |
| **Medications with potential effect on sleep,** n (%) | |
| Analgesics | 12 (17.4) |
| Psychoactive medications | 16 (23.3) |
| Sleep aids | 16 (23.2) |
| **Sleep Burden of COPD** | |
| Prevalence of Poor Sleep Quality, n = 64, PSQI score ≥ 5, n (%) | 55 (86) |
| Epworth's Sleep Scale, n = 64, mean ± SD | 8.0 ± 5.1 |
| **Neurocognitive Burden of COPD** | |
| **Geriatric Depression Scale**, n = 66, mean ± SD | 12.9 ± 7.6 |
| Normal (GDS score 0–10), n (%) | 21 (31.8) |
| Mild depression (GDS score 11–20), n(%) | 35 (53.0) |
| Severe depression (GDS score 21–30), n(%) | 10 (15.2) |
| **SLUMS Examination**, n = 67, median (IQR) | 25 (23–27) |
| Normal (Score 27–30), n (%) | 18 (26.9) |
| Mild Cognitive Impairment (Score 21–26), n (%) | 40 (59.7) |
| Dementia (Score <21), n (%) | 9 (13.4) |

**Abbreviations**: SD–Standard deviation, IQR–Interquartile range, BMI–Body mass index, FEV1%—Forced Expiratory Volume in first second (% of expected), GOLD–Global initiative for Obstructive lung disease, CAT score–COPD assessment test score, mMRC–modified Medical research council, SGRQ–St. Georges Respiratory Questionnaire, SLUMS–St. Luis University Mental State Examination.

less than 90% of sleep efficiency. Co-morbidities affecting sleep were also prevalent in our study population. Nearly 7 in 10 Veterans (68.2%) had depressive symptoms (GDS score >10). Similarly, 59.7% of the Veterans had mild cognitive impairments and 13.4% met SLUMS criteria for dementia. Both depression and cognitive impairments can affect sleep health. One quarters of the Veterans had diagnosed sleep apnea (17 of 69) and related disorder and all were treated. ESS, which measures general sleepiness, was 8.0 ± 5.1 (mean ± SD) slightly higher than expected in normal individuals at baseline (ESS in normal individuals, 5.9±2.2, range 2–10) [31].

Chronic pain was present in 52% and psychiatric conditions including PTSD were present in 48% of our study population. Despite the high prevalence of pain and psychiatric conditions, only 17.4% of our Veterans were taking pain medications including narcotics and only 23.3% were taking psychoactive medications including antipsychotics. About one-quarter (23.2%) were on sleep aids (either prescription or over-the-counter) and 26% were on home supplemental oxygen.

## Effect of PR on COPD related outcomes

Summary of effects of PR on COPD and sleep related variables are presented in Table 2. PR improved dyspnea and COPD related functional status and symptoms burden. SGRQ score significantly improved after PR (*p<0.0001*). PR was moderately efficacious in reducing COPD related respiratory symptoms (improvement in SGRQ by at least 8 points) in 50% of the Veterans. Dyspnea measured by the VAS part of the PFSDQ improved as well. The mMRC score improved on average by 0.76 points (95% CI 0.4–1.0; p<0.0001). However, only 29% (18/63) of the Veterans had a 2-point or more improvement in the mMRC score. The impact of COPD measured by CAT, decreased on average by 4.5 points (95% CI 2.5 to 6.5, p<0.0001). But only Veterans with high or very high impact of COPD at baseline (CAT score of >20) showed significant benefit. The 6-MWT also improved significantly after PR. On average, 6-MWD improved by 54m (95% CI 34m – 74m).

We observed statistical improvement in cognitive function by SLUMS score but only a trend toward improvement in mood (GDS score, median difference of 2.5, p = 0.051). The ESS did not change after PR in the entire cohort as well as when stratified by presence of absence of sleep disorders (stratified data not presented in the manuscript).

## Effect of PR on sleep

The effect of PR on sleep is summarized in Table 3. The TBT and TST decreased after PR although not significantly. The SOL, SE, WASO and nightly awakenings did not change following PR. Stratified analysis of all sleep variables by severity of COPD, BMI, mood, mental status, 6-MWD and SGRQ did not show significant improvement after PR. Overall, PR did not improve sleep when measured quantitatively by the actigraphy. Veterans who were not taking any psychoactive medications, however, had decrease in TBT (median, IQR: Pre-PR 532 min, 463–599 vs Post-PR 491 min, 426–544; p = 0.047) and TST (median, IQR: Pre-PR 418.5 min, 349–486 vs Post-PR 370 min, 314–465; p = 0.003) after PR. Similarly, Veterans who were not on any pain medications, TST decreased as well (median, IQR: Pre-PR 394 min, 337–459 vs 376 min, 322–439; p = 0.04).

Sleep quality measured by PSQI in this population did not improve after 8-weeks of PR. The average improvement in PSQI score in the entire cohort was 0.57 points (p = 0.07). In Veterans with poor sleep quality at baseline (pre-PR PSQI score of ≥ 5), however, PSQI score improved significantly (mean difference 0.79; p = 0.03) (Fig 2).

**Table 2. Effect of PR on COPD/respiratory, sleep and neurocognitive variables.**

| Variables | Pre-PR | Post-PR | Mean change (95% CI) | Significance (p-value) |
|---|---|---|---|---|
| **St. George's Respiratory Questionnaire (SGQR), n = 64** | | | | |
| SGQR score, mean ± SD | 60.9 ± 16.3 | 53.2 ± 16.4 | 7.7 (5.2 to 10.2) | <0.0001* |
| SGQR score, median (IQR) | 61.1 (48.8 to 72.52) | 56.7 (40.9 to 64.8) | | |
| **Pulmonary Functional Status & Dyspnea Questionnaire** | | | | |
| "Indicate how you have felt most days during past year?", n = 53 | | | | |
| VAS, mean ± SD | 5.6 ± 1.6 | 5.3 ± 1.7 | 0.3 (-0.1 to 0.7) | 0.15* |
| VAS score, median (IQR) | 6 (5 to 7) | 5 (4.2 to 6.2) | | |
| "Indicate how you feel today?", n = 60 | | | | |
| VAS, mean ± SD | 4.9 ± 2.2 | 4.0 ± 2.1 | 0.8 (0.2 to 1.5) | 0.01* |
| VAS Score, median (IQR) | 5 (4 to 6) | 4 (2 to 5.5) | | |
| "Indicate how you feel with most day-to-day activities?", n = 60 | | | | |
| VAS, mean ± SD | 5.9 ± 2.0 | 5.0 ± 1.9 | 0.9 (0.4 to 1.4) | 0.001* |
| VAS Score, median (IQR) | 6 (5 to 7) | 5 (4 to 6) | | |
| **COPD Assessment Test, n = 46** | | | | |
| CAT score, mean ± SD | 24.2 ± 7.0 | 17.5 ± 6.2 | 4.5 (2.5 to 6.5) | <0.0001** |
| CAT score, median (IQR) | 25 (20.5 to 29.5) | 21 (16 to 24) | | |
| **mMRC dyspnea scale, n = 63** | | | | |
| mMRC score, mean ± SD | 2.4 ± 1.1 | 1.6 ± 0.9 | 0.76 (0.4 to 1.0) | <0.0001** |
| mMRC score, median (IQR) | 2 (2–4) | 2 (1–2) | | |
| **SLUMS score, n = 64** | | | | |
| SLUMS score, mean ± SD | 24.5 ± 3.2 | 25.7 ± 2.8 | 1.2 (0.4 to 2.0) | 0.010* |
| SLUMS score, median (IQR) | 25 (23 to 27) | 26 (24 to 27) | | |
| **Geriatric Depression Scale, n = 66** | | | | |
| GDS score, mean ± SD | 12.7 ± 7.6 | 11.6 ± 7.7 | 1.1 (-0.1 to 2.3) | 0.051* |
| GDS score, median (IQR) | 13.5 (6 to 18) | 11 (5 to 16) | | |
| **6-min Walk test, n = 66** | | | | |
| 6-MWD, meters, mean ± SD | 340 ± 114 | 394 ± 129 | 54 (34 to 74) | <0.0001* |
| 6-MWD, meters, median (IQR) | 373 (300.3 to 431.9) | 407.3 (331.6 to 165.7) | | |
| **Epworth Sleepiness Scale, n = 61** | | | | |
| ESS score, mean ± SD | 7.7 ± 5.1 | 8.0 ± 4.5 | -0.3 (-1.4 to 0.7) | 0.54** |
| ESS score, median (IQR) | 8 (4 to 11) | 8 (5 to 11) | | |

**Abbreviations:** 95% CI (95% Confidence Interval); Statistics:

*Wilcoxson Signed rank test

**paired t-test.

A post-hoc analysis of actigraphic measure of sleep by quality of sleep at baseline (normal vs poor sleepers) was performed as well. There was no difference in quantitative sleep parameters in poor sleepers as compared to Veterans with normal sleep quality by PSQI.

## Discussion

In this retrospective study, the most important finding was a significant improvement in the subjective sleep quality after pulmonary rehabilitation in Veterans who had poor quality sleep before PR. We did not find improvements in objective measures of sleep by actigraphy despite improvements in dyspnea, CAT, SGRQ, and mood/cognitive scores along with improved exercise.

**Table 3. Sleep related primary and secondary outcomes after 8-weeks of pulmonary rehabilitation.**

| | Before Pulmonary Rehabilitation | After Pulmonary Rehabilitation | | |
|---|---|---|---|---|
| **Primary outcomes** | | | | |
| **Actigraphy Sleep Variables, n = 69** | **Median (IQR)** | **Median (IQR)** | **Significance** | |
| Total Time in Bed, TBT (min) | 499 (447 to 567) | 491 (437 to 548) | 0.40* | |
| Total Sleep Time, TST (min) | 384 (331 to 425) | 377 (321 to 448) | 0.14* | |
| Sleep Onset Latency, SOL (min) | 29.7 (15.4 to 38.1) | 29.6 (11.8 to 43.4) | 0.99* | |
| Sleep Efficiency, SE (%) | 80.6 (73.0 to 84.9) | 79.5 (72.5 to 85.9) | 0.32* | |
| Wakefulness After Sleep Onset, WASO (min) | 51.3 (37.2 to 70.6) | 56.3 (33.5 to 75.6) | 0.70* | |
| Number of nightly Awakenings | 33.4 (25.3 to 45) | 35.5 (22.3 to 46.5) | 0.77* | |
| **Secondary outcomes** | | | | |
| **Pittsburg Sleep Quality Index (PSQI)** | **Mean (SD)** | **Mean (SD)** | **Mean Difference (^95% CI)** | **Significance** |
| PSQI score–all Veterans, n = 61 | 8.1 (3.4) | 7.5 (3.2) | 0.57 (-0.05 to 1.2) | 0.07** |
| PSQI score in Veterans with normal sleep quality at baseline (PQSI ≤5), n = 9 | 3.2 (0.97) | 3.7 (2.0) | -0.44 (-2.3 to 1.4) | 0.59** |
| PSQI score in Veterans with poor sleep quality at baseline (PQSI >5), n = 52 | 8.9 (2.9) | 8.2 (2.9) | 0.79 (0.07 to 1.4) | 0.03** |

*Wilcoxson Signed rank test

**Paired t-test; ^95% CI– 95% Confidence Interval.

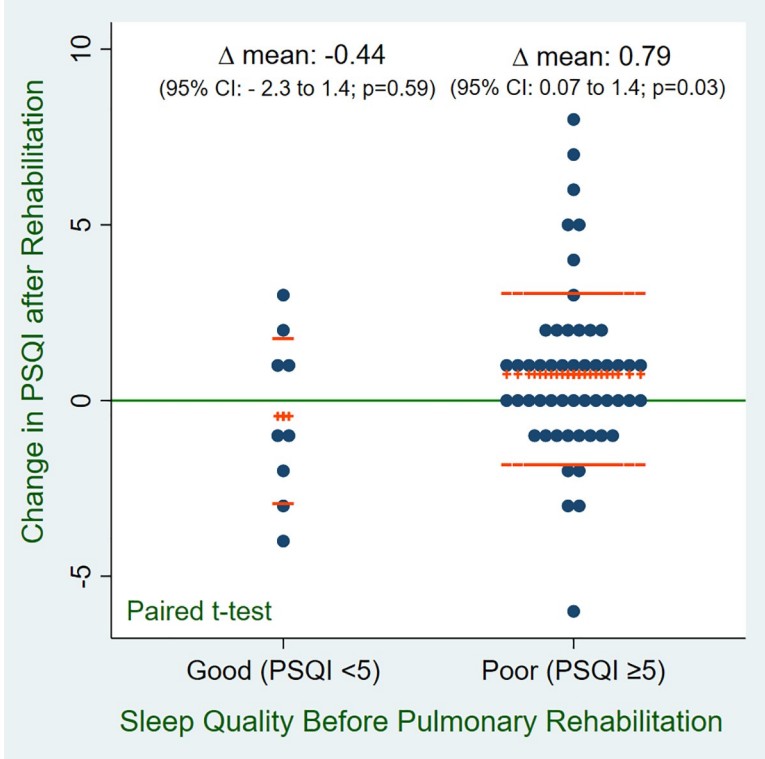

**Fig 2. Effect of PR in PSQI score in veterans with good sleep quality (PSQI <5) and poor sleep quality (PSQI ≥5).** Data are presented in mean change in PSQI among veterans with good sleep quality and poor sleep quality before and after PR. The line (+++) shows mean change and the line (—) shows data dispersion with ±1 standard deviation (SD). Abbreviations: PSQI, Pittsburg Sleep Quality Index; CI, Confidence interval.

Our cohort consistent of only Veterans and almost entirely males. Both factors can influence sleep. Veterans have high prevalence of psychiatric conditions such as PTSD, depression and anxiety as compared to general population [32]. Directly or indirectly these factors increase the prevalence of insomnia and poor-quality sleep. Gender affects sleep as well. Females have reported poor quality of sleep, longer sleep latency and increased nighttime awakenings as compared to males [33], however, when measured by PSG, females have better sleep efficiency, more time spent sleeping and less awakening [34]. In elderly population, this gender difference may be very small or non-significant and females have better correlation between objective and subjective sleep qualities [35]. All these studies are done in healthy volunteers, therefore, gender effect in people with COPD has not been studied.

The prevalence of poor-quality sleep in our cohort (86%) was higher than reported in previous studies [1, 2]. Quantitatively, our study cohort had poor sleep as well. For example, median sleep efficiency was 80.6% (IQR 73.0 to 84.9%). Although, PSQI poorly correlates with objective measures of sleep such as polysomnography or actigraphy [23], our cohort had poor sleep both qualitatively and quantitatively. The PR program was highly effective in several outcomes that were expected to improve. In theory, as well as qualitative data from prior studies (PSQI data) and robust improvement in COPD related variables with PR in our study, we expected improvement in objective sleep outcomes by actigraphy. However, the trend in our data was for a decrease in total bedtime, total sleep time and increased restless sleep, albeit without statistical significance. Some of the data trends in our study are in disagreement with recently published data from Cox et al. [36] but statistically similar. TST and SE increased in Cox et al. study but either decreased or remained unchanged in our study. Sleep parameters of our cohort were also slightly different from Cox et al. cohort. Our study cohort had markedly high total number of awakenings (median 33.5 vs 9) and SOL (median 29.7 vs 20) as compared to Cox et al. On the other hand, WASO was markedly higher in the Cox et al. cohort (median 79 vs 51.3). The Cox et al. study included both home-based and center-based PR participants but not all participants completed 8-weeks of PR. They also did not report subjective sleep measures. In contrast, we included only center-based participants and all participants completed a full 8-weeks of PR. We also reported both objective sleep measures (actigraphy data) and subjective sleep quality measures (PSQI data). To our knowledge this is the largest cohort of COPD patient undergoing PR and sleep evaluation both objectively and subjectively.

Subjective measures of sleep quality by PSQI have previously been studied in peoples with COPD undergoing PR. The results of those studies are mixed [1, 16, 37]. McDonnell et al. [1], in his 28-patient cohort, reported a mean difference in the PSQI score of 0.79 (95% CI -0.35 to 1.93) after PR. Soler et al. [16] included patients with both restrictive and obstructive lung diseases. In his cohort of 48-COPD patients, he reported significant improvement in the global PSQI index (6.6 ± 3.9 to 5.5 ± 3.6, p = 0.01). In these 48 patients, the proportion of patients with poor sleep quality decreased from 58% to 47% (p<0.001) [16]. In another study, Lan et al. [37], showed a significant decrease in the PSQI score (mean difference of 1.89) after PR in COPD patients. The number of patients with PSQI >5 decreased by 20.6% in that cohort (n = 34). In comparison to the above studies, we only found significant improvement in sleep quality in the subgroup of our Veterans who had poor quality sleep before the beginning of PR. While our study and some other studies, as discussed above, have found statistically significant improvement in PSQI but none of the study reached minimal clinically important difference (MCID) for PSQI which is 3 points. This underlines the fact that PR alone may not be sufficient to improve sleep. Interventions such as formal sleep education and cognitive behavioral therapy if incorporated in the PR program, may have more impact on sleep.

COPD impairs sleep quality potentially by sleep fragmentation from increased arousals and sleep disruptions. A multitude of factors may be involved including symptoms such as cough

and dyspnea and underlying disease pathology such as lung hyperinflation, hypercapnia and hypoxemia. Other factors may also contribute including COPD comorbidities of anxiety, depression, pain, and neuropsychiatric conditions. Lastly, the treatments of COPD including oxygen therapy, beta-agonist therapy and anticholinergic therapy likely disrupt sleep as well. The natural effect of sleep itself worsens symptoms of COPD by decreasing ventilation, increasing airflow obstruction leading to hypercapnia and hypoxemia. PR improves the overall symptom burden in COPD, improves QoL and also improves comorbid conditions such as anxiety, depression [38, 39] and cognitive status [40].

Exercise is one of the major components of PR, therefore, we expected improvement in sleep quality following PR. However, lack of significant improvement in sleep in COPD after 8-weeks of PR suggests a possible complex interaction between pathophysiology of COPD, COPD symptoms, associated comorbidities and sleep. We did not observe significant objective benefit in our cohort which may suggest that PR programs substantially longer than 8-weeks are required to demonstrate improvement in other measure of sleep quality and quantity. Lack of improvement in actigraphy measures may also be related to severity of COPD in our cohort.

Exercise affects sleep in several ways and these effects may vary depending on the duration and intensity of the exercise [41]. Acute exercise may improve total sleep time and slow wave sleep probably by affecting core body temperature, increasing growth hormone, decreasing insulin resistance and decreasing vagal tone. Regular exercise improves sleep by modulating inflammation, circadian rhythm and mood [41]. Data on the effects of sleep in the elderly population are limited and those studies mostly included healthy individuals. Available data suggest that at least 10 to 16 weeks of regular exercise in elderly population improves sleep quality, reduces sleep fragmentation and improve sleep efficiency [42–46]. Increased sleep continuity and sleep depth was noted in a polysomnography (PSG) study after 16 weeks of exercise [47]. But the same study failed to show improvement in sleep quality (PSQI), total sleep time and sleep efficiency. A systematic review by Yang et al. showed only improvement in sleep quality (PSQI and SOL) but not sleep duration, efficiency and disturbances [43]. Regular exercise over 12 months or more may be needed to observe objective sleep benefits [41].

Another notable finding in our cohort was that those Veterans who were not on any pain medications or psychoactive medications had decrease in sleep durations (TBT and TST) but no effect on sleep efficiency. This may illustrate the effect of pain and psychiatric condition as well as its treatment on sleep.

Objective measures of sleep such as actigraphy measure different aspects of sleep than the subjective PSQI. As such, actigraphy outcomes correlate poorly with PSQI [23, 48]. Aili et al. recommended at least six-nights of measurements as a reliable measure of sleep quality [48]. We discovered similar discrepancy between subjective and objective sleep measure.

## Limitations of the study

There are several limitations to our study. Some of the limitations are inherent to the retrospective design of the study such as lack of randomization and lack of a control group. However, a "before-and-after intervention" study design has the benefit of decreasing inter-subject variability. Multiple comparisons for the stratified analyses were done which may have decreased the power of the study. We only included participants with complete pre-PR and post-PR actigraphy data eliminating the problems of missing data. Another potential limitation of the study is the measurement of sleep variables with actigraphy. Actigraphy has acceptable validity and reliability [19, 20, 49] but has several limitations as compared to PSG. Actigraphy is one dimensional measurement of sleep based on movements and PGS gathers

sleep data from multiple inputs such as electroencephalogram, electro-oculogram and electro-myogram [49]. Therefore, certain sleep variables such as SOL is more accurate in PSG [50]. More definitive measurement of sleep by PSG would have been inconvenient and more time consuming for the subjects. An overnight PSG would have given only one day of sleep data. On other hand, actigraphy is able to gather data for multiple days providing a more robust picture of sleep disturbances than would be obtained from one night only [49]. Daytime activities and naps also affect sleep. Daytime napping can decrease sleep efficiency and duration of nocturnal sleep [51, 52]. On the other hand, people with COPD with better sleep have increased daytime physical activities [6]. We did not collect data on daytime activities or nap in our study cohort.

The study population was entirely composed of Veterans and nearly all were male. As discussed earlier, Veterans have high prevalence of post-traumatic stress disorder and other related psychiatric disorders, which, in turn, may cause a higher prevalence of poor sleep quality. Our PR program excludes current smokers as well. Therefore, our findings may not be generalizable to current smokers, females with COPD or the non-Veteran COPD population.

## Conclusions

In summary, we found a high prevalence of poor sleep quality in our study population undergoing PR. Quantitative measures of sleep by actigraphy did not improve after 8-weeks of PR. Qualitative measures of sleep by PSQI improved significantly in Veterans with poor sleep quality but the improvement was small and may not be clinically important. These findings suggest a complex relationship between COPD, sleep and exercise but also show benefit of PR in people with COPD with poor sleep quality.

## Acknowledgments

This material is the result of work supported with resources and the use of facilities at the VA Loma Linda Healthcare System, Loma Linda, CA.

All authors reviewed and approved final version of the draft manuscript and vouch for the accountability, accuracy and integrity of any part of the work.

## Author Contributions

**Conceptualization:** Suman B. Thapamagar, Kathleen Ellstrom, James D. Anholm, Ramiz A. Fargo, Nagamani Dandamudi.

**Data curation:** Suman B. Thapamagar, Kathleen Ellstrom, Nagamani Dandamudi.

**Formal analysis:** Suman B. Thapamagar, James D. Anholm.

**Methodology:** Suman B. Thapamagar, Kathleen Ellstrom, James D. Anholm, Ramiz A. Fargo, Nagamani Dandamudi.

**Project administration:** Suman B. Thapamagar, Kathleen Ellstrom, Ramiz A. Fargo, Nagamani Dandamudi.

**Resources:** Kathleen Ellstrom, James D. Anholm, Nagamani Dandamudi.

**Software:** Suman B. Thapamagar.

**Supervision:** Kathleen Ellstrom, James D. Anholm, Ramiz A. Fargo, Nagamani Dandamudi.

**Writing – original draft:** Suman B. Thapamagar, Kathleen Ellstrom, James D. Anholm.

**Writing – review & editing:** Suman B. Thapamagar, Kathleen Ellstrom, James D. Anholm, Ramiz A. Fargo, Nagamani Dandamudi.

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
