## [Decision Letter · Decision Letter 0]

15 Oct 2020

PONE-D-20-23683

Impact of Pulmonary Rehabilitation in Sleep in COPD patients measured by Actigraphy.

PLOS ONE

Dear Dr. Thapamagar,

Thank you for submitting your manuscript to PLOS ONE. After careful consideration, we feel that it has merit but does not fully meet PLOS ONE’s publication criteria as it currently stands. Therefore, we invite you to submit a revised version of the manuscript that addresses the points raised during the review process, as you can see below.

We look forward to receiving your revised manuscript.

Kind regards,

Federica Provini

Academic Editor

PLOS ONE

2. In your ethics statement in the manuscript and in the online submission form, please provide additional information about the patient records used in your retrospective study. Specifically, please ensure that you have discussed whether all data were fully anonymized before you accessed them.

4. Please remove your figures from within your manuscript file, leaving only the individual TIFF/EPS image files, uploaded separately.  These will be automatically included in the reviewers’ PDF.

**Comments to the Author**

1. Is the manuscript technically sound, and do the data support the conclusions?

Reviewer #1: Partly

Reviewer #2: Yes

2. Has the statistical analysis been performed appropriately and rigorously? 

Reviewer #1: I Don't Know

Reviewer #2: Yes

3. Have the authors made all data underlying the findings in their manuscript fully available?

Reviewer #1: Yes

Reviewer #2: Yes

4. Is the manuscript presented in an intelligible fashion and written in standard English?

Reviewer #1: Yes

Reviewer #2: Yes

5. Review Comments to the Author

Reviewer #1: Thank you for the opportunity to review this work which presents both subjective and objective sleep quality findings in a group of veterans with COPD who have undertaken pulmonary rehabilitation. The authors report findings that are largely consistent with the literature in this area, albeit a limited field of literature.

I have a couple of key queries to raise; followed by some more general comments.

- In the abstract and introduction the authors suggest that this is the first study in COPD to undertake actigraphic assessment of sleep relative to pulmonary rehabilitation. They then contradict this in the discussion with reference to the paper by Cox et al (reference 27). In the discussion the authors do accurately highlight theirs is the first study to include both subjective and objective assessment of sleep quality. Please amend all earlier statements to accurately reflect the work under consideration is not the first to use actigraphy in assessing sleep quality relative to pulmonary rehabilitation in COPD; but rather the first to combine both subjective and objective assessment of sleep quality.

- There are a number of discrepancies between the data in tables, data in text and associated text wording that tell a conflicting message. Depending on what is the correct data this may actually change the findings of the paper.

Pg 15: The text indicates that TBT and TST increased for veterans taking psychoactive medications, however the data for TST does not support this (pre median 394 minutes, post median 376 minutes). Please amend as appropriate. Conversely, in the discussion it is indicated that there is a decrease in TST. Similarly, the data on page 15 for veterans not taking psycho active medications indicates a reduction in TST (pre 418mins, post 370mins) but the text indicates an increase in TST.

- Table 3: the mean difference for change in PSQI for veterans with poor sleep quality needs to be a negative number ie. a decline in PSQI which corresponds to an improvement. Similarly for Figure 2.

Conversely, PSQI for veterans who did not have poor sleep quality needs to be a positive number ie. an increase in PSQI score which corresponds to a decline in sleep quality. Similarly for Figure 2.

- Table 2: Data for 6MWD suggests there was a significant decline in 6MWD post PR greater than the clinically meaningful difference (54m decline). The mean difference in the table needs to be presented as a negative number. This is also inaccurately reported in the text of the results as an improvement in 6MWD. Can the authors please check the data entered into the table in the event the pre and post rehab data have been transposed? Or amend the findings as presented. If, indeed, there was a decline in 6MWD this might suggest that in these participants PR did not achieve the expected outcomes/benefits associated with exercise capacity, and could be a contributing factor to the findings ie. insufficient training dose to see potential changes in sleep parameters?

The above queries may well result in relevant amendments throughout all sections of the manuscript.

Abstract:

- I’m not sure that ‘before-and-after’ requires hyphenation?

Introduction:

- The authors might consider describing the participants as ‘people with COPD’ as opposed to ‘patients with COPD’ as the study was not conducted in inpatients.

Methods:

- It is unclear how this study did not exclude current smokers, when the PR program under consideration does not include current smokers. Can the authors please clarify this wording.

- It would be helpful to indicate the session duration for the 3x PR sessions each week, and ideally what proportion of each class comprised exercise training.

- Did participants in this study also keep a sleep diary? If yes, how does patient documented timing of factors such as bed time line up with that produced by actigraphy?

- Can the authors please provide brief additional details as to how the actigraph device in question determines sleep factors, particularly bed-time vs sleep onset vs wake time vs get up time.

Results

- The sub heading ‘Prevalence of sleep…..’ does not seem to accurately reflect the content in this section. The authors may consider amending this sub-heading, possibly ‘Sleep quality and impacts on sleep’.

Discussion:

- Sentence one – please specify ‘subjective’ sleep quality improvement (if this is accurate relative to results queries)

- Can the authors comment on their PSQI findings, relative to the literature, in the context of the minimal important difference for the PSQI? While this and some studies have identified statistically significant improvements in PSQI with PR, no studies have demonstrated an improvement that exceeds the MID.

- The final paragraph of the discussion prior to limitations is a largely a repeat of already presented text and could be amended or deleted.

- It would add depth to this report if the authors specifically articulated the ‘disagreement in trends’ that they note between their work and that published by Cox et al. On reviewing that paper, it would appear that both studies have similar findings in that both studies found no change in sleep parameters measured by actigraphy following PR; which would be useful to articulate in terms of placing this study within the context of other findings.

Typographical errors:

- Pg 10, line 1: Typographical error. ‘Out’ PR program, should read ‘Our’ PR program

- Pg 12: Typographical error. ‘St Louise’ mental status exam, should read ‘St Louis’

- Pg 12. Patients clinical chart was ‘accessed’ by… (currently reads access)

- Pg 13 line 2 amend to read ‘clinically significant difference’

- Pg 14 – 6MWD has not been previously defined in the text of the manuscript

Reviewer #2: PONE-D-20-23683

Comments to the authors;

The study of Thapamagar and collaborators entitled "Impact of Pulmonary Rehabilitation in Sleep in COPD patients" is relevant as patients with COPD have significant sleep disorders, including poor sleep quality (the focus of this manuscript), obstructive sleep apnea, and nocturnal oxygen desaturations among others.

Increasing awareness about such sleep disturbances is relevant. Non-pharmacological approaches, such as pulmonary rehabilitation programs can be considered. Objective measurements of sleep quality can provide a scientific rationale to implement such therapies.

In this particular work, the authors tested the hypothesis that actigraphy would demonstrate improvement in objective sleep quality in patients with COPD attending a VA pulmonary rehabilitation program.

I think the manuscript could benefit from a revision. My comments are as:

1. The authors state that they did a retrospective study. They analyzed a large dataset of subjects that attended the PR program at the VA system with COPD. My understanding is that the subjects had the actigraphy (all of them) for sleep assessment as a routine part of the program? Or to assess daily activity? Please explain. Was it planned initially as a research study? If not, please expand on why your program is using Actiwatch on a PR program.

2. Expanding the discussion about male predominance is essential. The difference in sleep patterns from the gender perspective may help understand some of the results, and not limit the comment on the limitation section (which it should also be described as the authors did).

3. There is some discrepancy about the severity of the disease (please include 'severe and very severe' when describing the population -2 and 3- and the reference to GOLD severity guidelines). Why the patients had relatively preserved exercise tolerance measured by 6MW but are mostly on the 'severe' to 'very severe' category? That could happen, but less likely with such a high CAT score (mean of 24.8 +/- 7).

4. As very briefly discussed by the authors, pharmacology may have played an essential role in this population. However, I didn't find the multivariate analysis considering this variable as a whole ('any meds that may affect sleep). The impact of such medications on sleep architecture, expanding the discussion, and running the analysis may help.

5. COPD and sleep quality have been assessed (objectively)with PSG, contrary to what the authors describe. Discussing differences/limitations on actigraphy on sleep assessment seems important, contributing to the negative results.

6. The factors on the background influencing poor sleep (second paragraph) could use seminal references. The pharmacological effects and the reference (11) presented is not optimal.

7. The PR described (please add exercise tolerance on the third paragraph) also need some of the seminal or best references (Ries et al. in Annals, ATS PR document, or alike).

8. The paragraph starting (background section) 'there is also evidence that COPD…….. outcomes' will fit better in the discussion section.

9. Do not abbreviate Loma Linda VA (VALL) as it is not standard.

10. Typo on the last paragraph on methods' Out' needs to be 'Our', I guess.

11. Subheading explaining the actigraphy assessment. Briefly, what does, and how you use it (already described). Rephrase the 'use of the actigraphy' paragraph and put on the discussion. Also, it is simple to set, but not as simple to analyze for research purposes. A brief discussion of the actigraphy limitations seems necessary. "Actigraphy is a validated method to assess sleep, etc.'. It could be from Ancoli-Israel paper cited or equivalent.

12. There is no clarity on the overweight group (BMI 25-<30). Please expand on the discussion as the mean was 29? Table result (27.5) and text do not match BMI; please correct with the right number.

13. In the results section, limit to describe the % found. Avoid as much as possible 'nearly', majority, three-quarters by the %, etc. Just write the results.

14. SGRQ has an error/typo; 31.1 or 61? See text/table discrepancy.

15. mMRC is not expected to change that much. The authors seemed surprised by the results. Personal comment.

16. The description of the cognitive findings can be on the table and expand on the discussion. Those are not the primary or secondary aim for the study.

17. I would recommend running the analysis excluding patients on oxygen -26%- (improves sleep) and or obstructive sleep apnea (I assume there will be some). Both can interfere with the results. Despite having fewer subjects, may help better understand the results. It is very briefly discussed in two lines but a very general approach, and it is relevant for this study.

18. The results for PR are not necessary on the results section but a table and the discussion briefly (to demonstrate the program was effective). Are well known. Perhaps shortly name the ones that improved (quality of life, exercise tolerance, etc.).

19. The discussion could benefit from more profound thoughts from the authors regarding the benefits on the more severe patients (CAT score >20) as well as the ones with already low sleep quality (PSQI >5). Also, about the effects/impact of medication on sleep.

20. Some brief discussion comparing objective measurements of sleep would help the reader not familiar with sleep medicine (PSG vs. Actigraphy).

21. The paragraph starting with 'Exercise is one of the ….' Has to be before the previous one 'exercise affects sleep.'

22. On the limitations, some paragraphs will be perhaps better to move up to the discussion section; 'Actigraphy has reasonable validity ….'. Also, the phrase is not too scientific; maybe it can be rephrased.

23. Discussion about the patient population. This is relevant. And then briefly can be said in the limitation section as well.

24. Nice tables. Perhaps the median doesn't add much to the study description. The dyspnea questionnaire expanded is not needed in much detail as it is beyond the study's intent. The Epworth was collected and had to be part of sleep discussion and assessment and not standing alone.

25. Perhaps the reader could benefit from a brief explanation of how sleep quality is measured. Many readers are not familiar with sleep variables.

6. PLOS authors have the option to publish the peer review history of their article (what does this mean?). If published, this will include your full peer review and any attached files.

Reviewer #1: No

Reviewer #2: No

---

## [Author Response · Author response to Decision Letter 0]

8 Dec 2020

PONE-D-20-23683

Response to the Reviewer #1

Authors greatly appreciate the excellent review, comments, critics and recommendations by the reviewers. We have put forward our best effort to answer your concerns and comments and amended our manuscript as appropriate. We continue to value your critics and comments. 

Here are our responses:

Comment by the reviewer:

- In the abstract and introduction the authors suggest that this is the first study in COPD to undertake actigraphic assessment of sleep relative to pulmonary rehabilitation. They then contradict this in the discussion with reference to the paper by Cox et al (reference 27). In the discussion the authors do accurately highlight theirs is the first study to include both subjective and objective assessment of sleep quality. Please amend all earlier statements to accurately reflect the work under consideration is not the first to use actigraphy in assessing sleep quality relative to pulmonary rehabilitation in COPD; but rather the first to combine both subjective and objective assessment of sleep quality.

Response:

We really appreciate you pointing out this oversight. The matter of the fact is that when we conceived this study and during initial write up, there were no published studies which used actigraphy to assess sleep relative to pulmonary rehabilitation. While we were working on our data analyses, Cox et al. published their data. As suggested by the reviewer, we will amend our statement in the abstract and introduction section to reflect use of both subjective and objective assessment of sleep quality. 

Comment by the reviewer:

- There are a number of discrepancies between the data in tables, data in text and associated text wording that tell a conflicting message. Depending on what is the correct data this may actually change the findings of the paper.

Pg 15: The text indicates that TBT and TST increased for veterans taking psychoactive medications, however the data for TST does not support this (pre median 394 minutes, post median 376 minutes). Please amend as appropriate. Conversely, in the discussion it is indicated that there is a decrease in TST. Similarly, the data on page 15 for veterans not taking psycho active medications indicates a reduction in TST (pre 418mins, post 370mins) but the text indicates an increase in TST.

Response:

Authors greatly appreciate reviewer pointing out this discrepancy. In fact, TBT and TST data among Veterans not taking pain medications or psychoactive agents were inadvertently swapped and the same data was erroneously interpreted as increased rather than decreased. The data crossed checked and error was corrected. 

This segment of the manuscript reads as below before correction: (errors in red)

“Veterans who were not taking any pain medications, however, had an increase in TBT (median, IQR: Pre-PR - 481 min , 456-554 vs. Post-PR 491 min, 439 – 541; p=0.047) and TST (median, IQR: Pre-PR: 394 min, 337-459 vs Post-PR 376 min, 322 – 439; p=0.042) after PR. Similarly, Veterans who were not on any psychoactive medications, TST increased as well (median, IQR: Pre-PR: 418.5 min, 349-486 vs. Post-PR 370 min, 314 – 465; p=0.003)”

The updated/corrected paragraph will read as below: (corrections in blue)

“Veteran who were not taking any psychoactive medications, however, had decrease in TBT(median, IQR: Pre-PR 532 min, 463-599 vs Post-PR 491 min, 426-544; p=0.047) and TST (median, IQR” Pre-PR 418.5 min, 349 – 486 vs Post-PR 370 min, 314 – 465; p=0.003) after PR. Similarly, Veteran who were not on any pain medications, TST decreased as well (median, IQR: Pre-PR 394 min, 337 – 459 vs 376 min, 322 – 439; p=0.04).”

This portion of the data was not presented in the tables. If editorial team and reviewers think it will be useful, the authors are open to add to the table or authors will submit this section of data as a supplemental table.

Changes were made in the discussion section to reflect above mentioned changes. 

- Table 3: the mean difference for change in PSQI for veterans with poor sleep quality needs to be a negative number ie. a decline in PSQI which corresponds to an improvement. Similarly for Figure 2.

Conversely, PSQI for veterans who did not have poor sleep quality needs to be a positive number ie. an increase in PSQI score which corresponds to a decline in sleep quality. Similarly for Figure 2.

Response: We appreciate this discussion. It is reasonable to think that change in PSQI is negative in the language of numbers (ie PSQI decreased and therefore it should be marked as negative). However, the construction of the PSQI is such that a lower number represents less severe symptoms and improvement from prior. Since, we calculated all variables as difference from Pre-PR to Post-PR values, we adopted positive change for PSQI as improvement. If reviewers strongly feels that the convention is otherwise please advise. Authors are be happy to reconsider. 

- Table 2: Data for 6MWD suggests there was a significant decline in 6MWD post PR greater than the clinically meaningful difference (54m decline). The mean difference in the table needs to be presented as a negative number. This is also inaccurately reported in the text of the results as an improvement in 6MWD. Can the authors please check the data entered into the table in the event the pre and post rehab data have been transposed? Or amend the findings as presented. If, indeed, there was a decline in 6MWD this might suggest that in these participants PR did not achieve the expected outcomes/benefits associated with exercise capacity, and could be a contributing factor to the findings ie. Insufficient training dose to see potential changes in sleep parameters?

The above queries may well result in relevant amendments throughout all sections of the manuscript.

Response: We again appreciate catching this by our reviewer. In fact, the mean+/-SD data was transposition during the construction of the table. It has been re-checked and verified from the main data output sheet. No changes in the manuscript was needed except correction of the transposition of mean+/-SD values. 

Abstract:

- I’m not sure that ‘before-and-after’ requires hyphenation?

Response: Authors agree that it does not improve the sentence. Hence, we’ll remove the hyphenation.

Introduction:

- The authors might consider describing the participants as ‘people with COPD’ as opposed to ‘patients with COPD’ as the study was not conducted in inpatients.

Response: We agree with describing the participants as “people with COPD”. It reads better as well. “Patients” in most places were replaced by “people with COPD” as appropriate. In some places, where another article was referenced or the flow of the sentence was impeded by the phrase “people with COPD”, it was not changed. 

Methods:

- It is unclear how this study did not exclude current smokers, when the PR program under consideration does not include current smokers. Can the authors please clarify this wording.

Response: Our PR program excludes people with COPD who is a current smoker. As a pilot project, however, few people with COPD who were currently smoker were allowed. The pilot project was quickly discontinued due to high drop-out rate among smokers in the program. Since the number of participants who smoke were small, we decided to include them in our study. 

Method section has been amended to reflect the explanation. 

- It would be helpful to indicate the session duration for the 3x PR sessions each week, and ideally what proportion of each class comprised exercise training.

Response: Each PR session comprised of 30-60 min of educational session and 2-2.5 hours of exercise session with at-east 30 min of treadmill session tailored to each individual’s tolerance to exercise. 

A paragraph detailing this has been added to the manuscript. 

- Did participants in this study also keep a sleep diary? If yes, how does patient documented timing of factors such as bed time line up with that produced by actigraphy?

Response: Since this study was retrospective in design, the participants were not expected to keep a sleep diary. 

- Can the authors please provide brief additional details as to how the actigraph device in question determines sleep factors, particularly bed-time vs sleep onset vs wake time vs get up time.

Response: Actigraphy continuously monitors movement and activities of subjects when worn. By analyzing these movements using an algorithm, sleep variables can be estimated. We have added 2 lines with a new reference to summarize the actigraphy under “Assessments, Data Collection and Outcomes” section of the manuscript. 

Results

- The sub heading ‘Prevalence of sleep…..’ does not seem to accurately reflect the content in this section. The authors may consider amending this sub-heading, possibly ‘Sleep quality and impacts on sleep’.

Response: Our intention was to describe and report Sleep quality and condition affecting sleep. We agree that “Prevalence of sleep” may not describe the section well. We made changes as suggested with some modification. 

Discussion:

- Sentence one – please specify ‘subjective’ sleep quality improvement (if this is accurate relative to results queries)

Response: This is accurate information and ‘subjective” has been added to the sentences. Thank you again. 

- Can the authors comment on their PSQI findings, relative to the literature, in the context of the minimal important difference for the PSQI? While this and some studies have identified statistically significant improvements in PSQI with PR, no studies have demonstrated an improvement that exceeds the MID.

Response: Authors agree that studies have not found significant effect of PR on sleep quality by PSQI (at least MCID or better). Plausible explanation of this could be that multitude of factors affect sleep, especially in conditions like COPD. As discussed in the manuscript, several of factors affect sleep in COPD and impact of only PR in improving sleep may be small. Therefore, additional measures which may help to improve may be needed and incorporated in the PR to improve sleep in such population. Measures such as cognitive behavioral therapy for insomnia (CBT-i) could be beneficial. 

Change in manuscript: the following was added to the paragraph discussing PSQI in the discussion section. 

“While our study and some other studies, as discussed above, have found statistically significant improvement in PSQI but none of the study reached minimal clinically important difference (MCID) for PSQI which is 3. This underlines the fact that PR alone may not be sufficient to improve sleep. Interventions such as formal sleep education and cognitive behavioral therapy if incorporated in the PR program, may have more impact on sleep.”

- The final paragraph of the discussion prior to limitations is a largely a repeat of already presented text and could be amended or deleted.

Response: The sentence, which was similar from prior paragraph has been deleted.

There are no studies assessing sleep in both objective and subjective methods in a COPD cohort. Our study is the first to report such data.

- It would add depth to this report if the authors specifically articulated the ‘disagreement in trends’ that they note between their work and that published by Cox et al. On reviewing that paper, it would appear that both studies have similar findings in that both studies found no change in sleep parameters measured by actigraphy following PR; which would be useful to articulate in terms of placing this study within the context of other findings.

Response: Authors agree with your statement that the findings are similar statistically. However, there are some differences in the actigraphic sleep architecture and change following PR between Cox et al. cohort and ours. For example, TST increased in Cox et al but decreased in ours. SOL and Awakenings were low in Cox et al. study but WASO was higher than ours. 

Few sentences detailing similarities and differences has been added to the paragraph. 

Typographical errors:

- Pg 10, line 1: Typographical error. ‘Out’ PR program, should read ‘Our’ PR program

- Pg 12: Typographical error. ‘St Louise’ mental status exam, should read ‘St Louis’

- Pg 12. Patients clinical chart was ‘accessed’ by… (currently reads access)

- Pg 13 line 2 amend to read ‘clinically significant difference’

- Pg 14 – 6MWD has not been previously defined in the text of the manuscript

Response: Thank you again for spotting all these typographical errors. 

Page 10, Line 1: Changed to ‘Our’

Page 12: Changed to ‘St Louis’

Page 12: Changed to ‘accessed’

Page 13: Changed to ‘clinically significant difference’

Page 14: We agree that 6-MWD was not previously defined in the manuscript. Added six minute walk distance before 6-MWD. 

PONE-D-20-23683

Response to Reviewer # 2

Authors greatly appreciate the excellent review, comments, critics and recommendations by the reviewers. We have put forward our best effort to answer your concerns and comments and amended our manuscript as appropriate. We continue to value your critics and comments. 

Here are our responses:

1. The authors state that they did a retrospective study. They analyzed a large dataset of subjects that attended the PR program at the VA system with COPD. My understanding is that the subjects had the actigraphy (all of them) for sleep assessment as a routine part of the program? Or to assess daily activity? Please explain. Was it planned initially as a research study? If not, please expand on why your program is using Actiwatch on a PR program.

Response: During early days of PR program at Loma Linda VA, a pilot study was done to incorporate activity tracker to track activity and sleep. The study included only few patients. An abstract was presented to ATS. However, the practice to use activity tracker continued as a standard of care for PR program at our institution.

No changes in manuscript made in response to this comment. Please advise if it is absolutely necessary amend the manuscript. 

2. Expanding the discussion about male predominance is essential. The difference in sleep patterns from the gender perspective may help understand some of the results, and not limit the comment on the limitation section (which it should also be described as the authors did).

 Response: added a paragraph:

Gender affects sleep as well. Females have reported poor quality of sleep, longer sleep latency and increased nighttime awakenings as compared to males[32], however, when measured by PSG, females have better sleep efficiency, more time spent sleeping and less awakening[33]. In elderly population, this gender difference may be very small or non-significant and females have better correlation between objective and subjective sleep qualities[34]. All these studies are done in healthy volunteers, therefore, gender effect in people with COPD has not been studied.

3. There is some discrepancy about the severity of the disease (please include 'severe and very severe' when describing the population -2 and 3- and the reference to GOLD severity guidelines). Why the patients had relatively preserved exercise tolerance measured by 6MW but are mostly on the 'severe' to 'very severe' category? That could happen, but less likely with such a high CAT score (mean of 24.8 +/- 7).

Response: Thank for pointing out the discrepancy. We added “very severe” in the sentence to reflect the accurate description of the severity of the disease. 

We were also intrigued by the finding that exercise capacity was relatively preserved in our study population with severe or very severe COPD with very high CAT score. In my clinical practice, I obtain mMRC and CAT score on every patient on every visit. I also get 6MWT if there is any decline in functioning. For some reason, I have seen large discrepancy in mMRC and CAT score. I do not have large sample to cross validate this discrepancy with 6MWT in my practice. However, we know that there is a very good correlation between mMRC and CAT score (r=0.788, p<0.0001; Kaltsakas, ERJ 2013). We also know that dyspnea scales correlate with 6MWT. If I have to guess, I would suspect that it has to do something about discrepancy between various components of CAT score vs mMRC, especially components of CAT score asking about dyspnea. This is something I intend to pursue in a separate project. It is also difficult to evaluate the relative preservation of exercise in our population from this data set because of the nature of the study (we are studying 6MWT in a population who is expected to or are exercising). 

4. As very briefly discussed by the authors, pharmacology may have played an essential role in this population. However, I didn't find the multivariate analysis considering this variable as a whole ('any meds that may affect sleep). The impact of such medications on sleep architecture, expanding the discussion, and running the analysis may help.

Response: This is very interesting aspect of COPD and sleep management and an intriguing question as well. We did consider COPD medications as a factor in our protocol and data analysis. We collected data on each category of the medications including short acting bronchodilators, LABA, LAMA and ICS. We also included other drugs affecting sleep such as sleep aids, psychoactive medications as well as pain medications. Nearly all participants were on short-acting bronchodilators (SABD) and nearly all were in some combination of LABA, LAMA or both. The difference in the pattern of SABD, LABA, LAMA and combination of those would be very small. Therefore, to isolate the effect of those medications on sleep on during and after PR, we would have required much larger sample. We did not track changes in COPD medication use pattern. Our PR program only admits patients after their medications were optimized and therefore, we do not expect large changes in COPD medications. To answer the effect of COPD medications on sleep and PR, a well-designed prospective study may be needed. For this reason, we did not even present data on COPD medications. In regards to all medications affecting sleep, we presented data on psychoactive medications and pain medications. Small proportions of participants were taking medications affecting sleep (less than 25% in all medication group affecting sleep).

No changes were made in response to this comment. 

5. COPD and sleep quality have been assessed (objectively) with PSG, contrary to what the authors describe. Discussing differences/limitations on actigraphy on sleep assessment seems important, contributing to the negative results.

Response: We agree that Sleep in people with COPD has been objectively assessed by PSG. If the reviewer’s comment arise from the last paragraph of the “introduction” segment, we merely wanted to point out that there are paucity of data objectively evaluating effect of PR in sleep in people with COPD not in COPD population as such. 

No changes were made to the manuscript in response to this comment. 

6. The factors on the background influencing poor sleep (second paragraph) could use seminal references. The pharmacological effects and the reference (11) presented is not optimal.

Response: We agree seminal references would help the paper. We have added a reference to improve this. 

McCarthy, B., et al., Pulmonary rehabilitation for chronic obstructive pulmonary disease. Cochrane Database of Systematic Reviews, 2015(2).

Spruit, M.A., et al., An official American Thoracic Society/European Respiratory Society statement: key concepts and advances in pulmonary rehabilitation. Am J Respir Crit Care Med, 2013. 188(8): p. e13-64.

Ries, A.L., et al., Pulmonary Rehabilitation: Joint ACCP/AACVPR Evidence-Based Clinical Practice Guidelines. Chest, 2007. 131(5 Suppl): p. 4S-42S.

We retained ref#11 for pharmacological effects since adding reference for individual agents would have been long. 

7. The PR described (please add exercise tolerance on the third paragraph) also need some of the seminal or best references (Ries et al. in Annals, ATS PR document, or alike).

Response: Added exercise tolerance as recommended. Also added the following reference:

Ries, A.L., et al., Pulmonary Rehabilitation: Joint ACCP/AACVPR Evidence-Based Clinical Practice Guidelines. Chest, 2007. 131(5 Suppl): p. 4S-42S.

8. The paragraph starting (background section) 'there is also evidence that COPD…….. outcomes' will fit better in the discussion section.

Response: Thank you for your careful review. If it is not prohibit, authors would like to keep the paragraph where it is. We believe it support the rationale why we wanted to do the study. We agree that it can fit well in the discussion section as well.

9. Do not abbreviate Loma Linda VA (VALL) as it is not standard.

Response: VALL has been removed and replaced with ‘VA Loma Lima Healthcare system’ or ‘our’ in the appropriate

10. Typo on the last paragraph on methods' Out' needs to be 'Our', I guess.

Response: Thank you for spotting the typographical error. I hunted down all “out” masquerading as “our”

11. Subheading explaining the actigraphy assessment. Briefly, what does, and how you use it (already described). Rephrase the 'use of the actigraphy' paragraph and put on the discussion. Also, it is simple to set, but not as simple to analyze for research purposes. A brief discussion of the actigraphy limitations seems necessary. "Actigraphy is a validated method to assess sleep, etc.'. It could be from Ancoli-Israel paper cited or equivalent.

Response: Added few sentences divided between method section and limitations. Hopefully it suffices.

An actigraph is small simple device typically worn on the wrist like a watch. It can, however, be worn on an ankle or waist. This device is a sophisticated movement sensor that continuously records movement and integrates it over time. Mathematical algorithms are then utilized to estimate sleep and wakefulness time [17].

“Actigraphy has acceptable validity and reliability [19, 20, 49] but has several limitations as compared to PSG. Actigraphy is one dimensional measurement of sleep based on movements and PGS gathers sleep data from multiple inputs such as electroencephalogram, electro-oculogram and electromyogram[49]. Therefore, certain sleep variables such as SOL is more accurate in PSG[50].”

12. There is no clarity on the overweight group (BMI 25-<30). Please expand on the discussion as the mean was 29? Table result (27.5) and text do not match BMI; please correct with the right number.

Response: BMI group of less than 30, 30 to 39 and 40 or more was chosen for simplicity since half of the subjects in less than 30 BMI group had normal BMI. There was no significant difference between 3 BMI group (as above) or 4 BMI group (including normal, overweight, obese and morbidly obese) in regards to outcomes. 

The mean BMI was 29 and median BMI was 27.5. Mean+/-SD presented in the abstract was changed to median BMI with IQR.

13. In the results section, limit to describe the % found. Avoid as much as possible 'nearly', majority, three-quarters by the %, etc. Just write the results.

 Response: Those phrase were replaced by numbers/data where appropriate.

14. SGRQ has an error/typo; 31.1 or 61? See text/table discrepancy.

Response: Thank you for spotting this typographical error as well. I cross-checked and verified that the true value is 61.1. It was changed in the manuscript. 

15. mMRC is not expected to change that much. The authors seemed surprised by the results. Personal comment.

Response: Thank you for sharing your experience and fact about mMRC. The truth is that I was surprised by the degree of change in mMRC. 

16. The description of the cognitive findings can be on the table and expand on the discussion. Those are not the primary or secondary aim for the study.

Response: Stratified analysis by cognitive function and depression did not show any significant change in both objective and subjective sleep measures. Intuitively we expected some effect of mood or memory on sleep but we found none. Hence, we did not discuss futher. We have been collecting further data on short term and long term cognitive change after PR. We plan to discuss this issue further on next paper. 

17. I would recommend running the analysis excluding patients on oxygen -26%- (improves sleep) and or obstructive sleep apnea (I assume there will be some). Both can interfere with the results. Despite having fewer subjects, may help better understand the results. It is very briefly discussed in two lines but a very general approach, and it is relevant for this study.

Response: We did perform stratified analysis based on oxygen use. Among people who were not using supplemental oxygen for chronic respiratory failure, there was no significant effect of PR on sleep (Actigraphy variables as well as PSQI). 

Among people who used supplemental oxygen (n=18), total sleep time (TST) decreased marginally (pre-PR TST: 376±71 minutes, mean±SD, vs post-PR 321±100 minutes; p=0.02). Also, sleep efficiency improved marginally (pre-PR sleep efficiency: 76.0±8.4% vs post-PR sleep efficiency: 70.7±11.8%; p=0.03). Since the sample size was small, we did not report in the manuscript. Should the editor or reviewer believe that this information is significant for the manuscript, authors are ready to include and discuss in the full manuscript. 

Similarly, stratified analysis of sleep variables (PSQI as well as actigraphy parameters) among people with or without sleep disorder, did not show any significant differences. 

18. The results for PR are not necessary on the results section but a table and the discussion briefly (to demonstrate the program was effective). Are well known. Perhaps shortly name the ones that improved (quality of life, exercise tolerance, etc.).

Response: Authors belief that it improves the flow of reading. If it not prohibiti, authors would like to keep the paragraph in the Effect of PR on COPD related outcomes subheading. 

19. The discussion could benefit from more profound thoughts from the authors regarding the benefits on the more severe patients (CAT score >20) as well as the ones with already low sleep quality (PSQI >5). Also, about the effects/impact of medication on sleep.

Response: In theory, we expected robust improvement in sleep variables in severe COPD patients (CAT score >20) and PSQI> 5. Stratified analysis did not show any benefit in sleep variables despite significant improvement in CAT score (among severe patient, CAT score >20: mean change – 6.32 points, 95% CI -4.4 to -8.3, p<0.001; not presented in the manuscript) after PR. Therefore, we did not want to speculate the effect or propose a theory based on our retrospective data. 

20. Some brief discussion comparing objective measurements of sleep would help the reader not familiar with sleep medicine (PSG vs. Actigraphy).

Response: Added few sentences on Actigraphy in the method section and comparison in the limitation section. Details on PSG would have been long and confusing since none of our outcomes are based on PSG data.

21. The paragraph starting with 'Exercise is one of the ….' Has to be before the previous one 'exercise affects sleep.'

 Response: Moved as recommended.

22. On the limitations, some paragraphs will be perhaps better to move up to the discussion section; 'Actigraphy has reasonable validity ….'. Also, the phrase is not too scientific; maybe it can be rephrased.

Response: Thank you for picking that up as well. I agree “reasonable” may be less scientific. However, society practice guidelines has described actigraphy as “acceptable”. Hence, will change to “acceptable” from “reasonable”

Morgenthaler, T., et al., Practice parameters for the use of actigraphy in the assessment of sleep and sleep disorders: an update for 2007. Sleep, 2007. 30(4): p. 519-29.

Recommendations: Actigraphy provides an acceptably accurate estimate of sleep patterns in normal, healthy adult populations and inpatients suspected of certain sleep disorders.

23. Discussion about the patient population. This is relevant. And then briefly can be said in the limitation section as well.

 Response: Added a paragraph:

Our cohort consistent of only Veterans and almost entirely males. Both of these factors can influence sleep. Veterans have high prevalence of psychiatric conditions such as PTSD, depression and anxiety as compared to general population [31]. Directly or indirectly these factors increase the prevalence of insomnia and poor quality sleep.

24. Nice tables. Perhaps the median doesn't add much to the study description. The dyspnea questionnaire expanded is not needed in much detail as it is beyond the study's intent. The Epworth was collected and had to be part of sleep discussion and assessment and not standing alone.

Response: Nearly half of the variable were normally distributed and rest were not. Representing data in both means+/-SD and median+/-IQR provides some symmetry. Also, expanded PFSDQ VAS score provides an additional measure for dyspnea. Authors would like to keep those measure if acceptable to the reviewers. If the reviewers truly believe that those should be truncated or removed, we will be happy to make changes. 

In regards to ESS, added few sentences in the result section

ESS, which measures general sleepiness, was 8.0±5.1 (mean±SD) slightly higher than expected in normal individuals at baseline (ESS in normal individuals, 5.9±2.2, range 2-10)

The ESS did not change after PR in the entire cohort as well as when stratified by presence of absence of sleep disorders (stratified data not presented in the manuscript).

Since ESS was slightly above normal range and did not change in the overall cohort and in stratified analysis, we decided not to discuss it further. 

Also, correction was made to the SD values for ESS in table 2 (the table contained SE not SD)

25. Perhaps the reader could benefit from a brief explanation of how sleep quality is measured. Many readers are not familiar with sleep variables.

Response: I did not quite understand how to respond to this comment. I added a small paragraph on PSQI in the method section. Discussion of all sleep variables would be too lengthy. I hope this suffices. 

 The added paragraph reads as below:

PSQI measures sleep disturbances and sleep habits during last one month period. The questionnaire has 19 individual items and includes open ended questions and Likert-like items. It generates seven clinically derived domains or components of scores which includes: sleep quality, sleep latency, sleep duration, habitual sleep efficiency, sleep disturbances, use of sleeping medications, and daytime dysfunction. The global PSQI score is calculated by adding scores from all seven domains {Buysse, 1989}.

---

## [Decision Letter · Decision Letter 1]

13 Jan 2021

PONE-D-20-23683R1

Impact of Pulmonary Rehabilitation in Sleep in COPD patients measured by Actigraphy.

PLOS ONE

Dear Dr. Thapamagar,

Thank you for submitting your manuscript to PLOS ONE. After careful consideration, we feel that it has merit but does not fully meet PLOS ONE’s publication criteria as it currently stands. Therefore, we invite you to submit a revised version of the manuscript that addresses the points raised during the review process.

We look forward to receiving your revised manuscript.

Kind regards,

Federica Provini

Academic Editor

PLOS ONE

There are a number of discrepancies between data in tables and data in text.

For example

1) in the text: ”Seventy five percentage of the study population had a high or very high impact of COPD on their daily lives with a mean CAT score of 24.4 ± 7.1

2) in the table: CAT score, n=52, mean ± SD 24.8 ± 7.1

If you want to report data on different subgroups, please clarify in the text or in the tables

Please, carefully check and resubmit again

---

## [Author Response · Author response to Decision Letter 1]

23 Jan 2021

PONE-D-20-23683

Response to the reviewer

Comment from the reviewer:

There are a number of discrepancies between data in tables and data in text.

For example

1) in the text: ”Seventy five percentage of the study population had a high or very high impact of COPD on their daily lives with a mean CAT score of 24.4 ± 7.1

2) in the table: CAT score, n=52, mean ± SD 24.8 ± 7.1

Response:

Thank you very much for spotting the typographical error. The value should be 24.8 ± 7.1 in the text as well (as in the table). 

 After re-reading the text again, I felt that the construct of the sentence was confusing. It was suggesting as if the mean CAT score presented in the text was for “High and Very high impact” group, which was not the intention. Therefore, we decided to make changes to 2 sentences in the baseline characteristic paragraph. 

Before change:

Nearly 80% of the Veterans had severe or very severe COPD (GOLD stage 3 or 4). Seventy five percentage of the study population had a high or very high impact of COPD on their daily lives with a mean CAT score of 24.4 ± 7.1.

The new sentences read as below:

Nearly 80% of the Veterans had severe or very severe COPD (GOLD stage 3 or 4) and the impact of COPD in their lives as measured by COPD Assessment Test (mean CAT score 24.8 ± 7.1) was very high. Moreover, 75% of the study population had a high or very high impact of COPD on their daily lives. 

The break down for CAT score groups was as follows:

 Medium impact group (CAT score >10 to 20)

 n=13, mean ± SD, 15.5 ± 3.0

 High or very high impact group (CAT score of > 20)

 n=39, mean ± SD: 27.9 ± 4.9

Proportional break down of the CAT score provides a good insight into the data. Adding additional piece of data would have cluttered the table more (in our opinion). Therefore, we decided not to add anything more to the table.

---

## [Editor Report · Decision Letter 2]

1 Feb 2021

PONE-D-20-23683R2

Impact of Pulmonary Rehabilitation in Sleep in COPD patients measured by Actigraphy.

PLOS ONE

Dear Dr. Thapamagar,

Please check carefully again the data in the text and in the tables because some disagreeents are still present.

For example- Non-obese pts: 52.3 % in the text and 62.3% in the table. 

We look forward to receiving your revised manuscript.

Kind regards,

Federica Provini

Academic Editor

PLOS ONE

---

## [Author Response · Author response to Decision Letter 2]

26 Feb 2021

PONE-D-20-23683

Response to the reviewer

Comment from the reviewer:

Please check carefully again the data in the text and in the tables because some disagreements are still present.

For example- Non-obese pts: 52.3 % in the text and 62.3% in the table. 

Response:

Thank you very much for spotting the typographical error. The value in the table (62.3%) is accurate. This was crossed check with data output and corrected in the updated manuscript.

We also carefully checked the entire manuscript for other typographical and formatting errors as well as grammatical errors. We found few and have made changes as below:

1) Results/Demographics: “80%” should have been “70%”

 Used to read: “Nearly 80% of the Veterans”

 Now reads: More than two-thirds (69.5%) of the Veterans

2) “GOLD stages 3 or 4” were changed to “GOLD stages III or IV”

3) Table 1: “%” signs inside the parentheses in the second column were removed since it was already indicated in the first column. For example:

Used to read as: 43 (62.3%)

Now reads: 43 (62.3)

4) Table 1: Minimal cognitive impairment was changed to Mild cognitive impairment since SLUMS does not have minimal cognitive impairment in the scale. 

5) All sentences with “veterans” were changed with “Veterans”

6) Result/Sleep quality and factors affecting sleep:

- Chronic pain prevalence was 52% not 42%. It was cross checked and corrected in the edited manuscript.

7) Results/Effect of PR on COPD related outcomes:

 -p value for change in CAT score was <0.0001 not <0.001.

8) Punctuations and grammatical errors were corrected as appropriate.

---

## [Editor Report · Decision Letter 3]

1 Mar 2021

Impact of Pulmonary Rehabilitation in Sleep in COPD patients measured by Actigraphy.

PONE-D-20-23683R3

Dear Dr. Thapamagar,

We’re pleased to inform you that your manuscript has been judged scientifically suitable for publication and will be formally accepted for publication once it meets all outstanding technical requirements.

Kind regards,

Federica Provini

Academic Editor

PLOS ONE

---

## [Editor Report · Acceptance letter]

5 Mar 2021

PONE-D-20-23683R3 

Impact of pulmonary rehabilitation in sleep in COPD patients measured by actigraphy 

Dear Dr. Thapamagar:

I'm pleased to inform you that your manuscript has been deemed suitable for publication in PLOS ONE. Congratulations! Your manuscript is now with our production department. 

Kind regards, 

on behalf of

Dr. Federica Provini 

Academic Editor

PLOS ONE